# The Bioaugmentation of Electroactive Microorganisms Enhances Anaerobic Digestion

Zheng-Kai An [1,2] , Young-Chae Song [1,2,*] , Keug-Tae Kim [3] , Chae-Young Lee [4] , Seong-Ho Jang [5] and Byung-Uk Bae [6]

1 Department of Environmental Engineering, Korea Maritime and Ocean University, Busan 49112, Republic of Korea; qgd2016azk@126.com
2 Interdisciplinary Major of Ocean Renewable Energy Engineering, Busan 49112, Republic of Korea
3 Department of Biological and Environmental Science, Dongguk University, Gyeonggi 10326, Republic of Korea; kkt38@dongguk.edu
4 Division of Civil, Environmental and Energy Engineering, The University of Suwon, Gyeonggi 18323, Republic of Korea; chaeyoung@suwon.ac.kr
5 Department of Bioenvironmental Energy, Pusan National University, Miryang 50463, Republic of Korea; jangsh@pusan.ac.kr
6 Department of Civil and Environmental Engineering, Daejeon University, Daejeon 34520, Republic of Korea; baebu@dju.ac.kr
* Correspondence: soyc@kmou.ac.kr

**Abstract:** Direct interspecies electron transfer (DIET) between electroactive microorganisms (EAMs) offers significant potential to enhance methane production, necessitating research for its practical implementation. This study investigated enhanced methane production through DIET in an anaerobic digester bio-augmented with EAMs. A horizontal anaerobic digester (HAD) operated for 430 days as a testbed to validate the benefits of bioaugmentation with EAMs. Anaerobic digestate slurry, discharged from the HAD, was enriched with EAMs in a bioelectrochemical auxiliary reactor (BEAR) under an electric field. This slurry enriched with EAMs was then recirculated into the HAD. Results showed bio-augmentation with EAMs led to an increase in volatile solids removal from 56.2% to 77.5%, methane production rate from 0.59 to 1.00 L/L.d, methane yield from 0.26 to 0.34 L/g CODr, and biogas methane content from 59.9% to 71.6%. It suggests that bio-augmentation enhances DIET, promoting the conversion of volatile fatty acids to methane and enhancing resilience against kinetic imbalances. The enrichment of EAMs reached optimal efficacy under an electric field intensity of 2.07 V/cm with a mean exposure time of 2.53 days to the electric field in the BEAR. Bio-augmentation with externally enriched EAMs is a feasible and effective strategy to optimize anaerobic digestion processes.

**Keywords:** electric field; bioelectrochemical reactor; direct interspecies electron transfer; resilience of anaerobic digestion; methanogenesis promotion

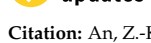



## 1. Introduction

Anaerobic digestion stabilizes organic matter and produces methane, but still faces unresolved inherent challenges. In anaerobic digestion, organic matter undergoes enzymatic hydrolysis, converting it to monomers, then is fermented into volatile fatty acids (VFAs) by acidogenic bacteria. These VFAs are subsequently further fermented into acetic acid and hydrogen. Finally, methanogenic archaea either dismutate acetate or reduce carbon dioxide with hydrogen to produce methane. Notably, the acetogenesis of VFAs is thermodynamically unfavorable [1,2]. Acetogenesis becomes feasible only when hydrogenotrophic methanogenesis maintains a low hydrogen partial pressure. The intrinsic issues in anaerobic digestion are closely tied to the physiological characteristics of methanogenic archaea, which are more susceptible to environmental variations than acidogenic bacteria [3–5].

Consequently, fluctuations in hydraulic loading rate, organic loading rate, and other environmental conditions can hinder methanogenesis, accumulating the intermediates, including volatile fatty acids (VFAs) and hydrogen, thereby inhibiting methanogenesis [6]. Furthermore, there are inherent losses in the enzymatic electron transfer from organic matter to methane, resulting in the methane yield generally falling short of the theoretical value [3,7].

Various attempts have been made to mitigate the challenges above in anaerobic digestion, primarily by operating at low hydraulic or organic loading rates or increasing biomass retention in the anaerobic digester [8–10]. However, these approaches alone have not fully addressed the inherent issues of anaerobic digestion. Recently, the potential of microbial electrolysis cells (MECs) in improving anaerobic digestion performance has received considerable attention [3,11,12]. In the MECs, electrochemically active microorganisms (EAMs) facilitate methanogenesis through direct interspecies electron transfer (DIET) mediated by polarized bioelectrodes [5,10,13]. EAMs encompass microbial species equipped with conductive proteins, including conductive pili or over-expressed cytochrome C to the outer membrane of the microbial cell [14,15]. EAMs involved in the methanogenesis through DIET include exoelectrogenic bacteria (EEB) and electrotrophic methanogenic archaea (EMA) [3,7]. Within MECs, EEBs can thrive on the anode surface, directly transferring electrons from organic matter to the polarized anode. EMAs can accept electrons from the cathode, facilitating methane production by reducing carbon dioxide. MECs offer a promising solution to address the unresolved issues of anaerobic digestion [5,10,13]. Nevertheless, several challenges related to electrode installation and maintenance have hindered the practical application of MECs in full-scale anaerobic digesters [7,16,17]. Hence, despite the significant potential of EAMs to enhance methanogenesis through DIET, the question of how to apply EAMs to anaerobic digestion has remained unanswered.

Fortunately, it has been discovered that an electric field can enrich EAMs within the bulk solution of anaerobic digesters [7,18]. EEBs in the bulk solution could be electrically connected to EMAs to promote methane production through DIET [3,7]. This suggests that the performance of conventional anaerobic digesters could be improved by enriching EAMs in the bulk solution. However, the crucial issue lies in enriching EAMs within the bulk solution. EAMs can be enriched in the bulk solution by installing polarized electrodes directly in the anaerobic digester to establish an electric field. However, several unresolved challenges exist in installing and maintaining the electrodes within anaerobic digesters [7,19]. For instance, installing electrodes requires a significant initial investment, and maintaining them necessitates ongoing costs. In particular, electrodes placed within the anaerobic digester can disrupt the agitation and cleaning processes [7,16,20]. Therefore, there is a need for further discussion on strategies for enriching EAMs in the bulk solution of conventional anaerobic digesters. This study proposes that EAMs could be enriched in an external small bioelectrochemical reactor without the severe issues associated with the electrode and bio-augmented to improve anaerobic digestion.

This study aimed to answer how to enrich EAMs in the bulk solution and how to apply them to improve conventional anaerobic digestion. A lab-scale horizontal anaerobic digester (HAD), operated in a conventional mode, was used for the experiment. The anaerobic digestate was discharged from the HAD and injected into a bioelectrochemical auxiliary reactor (BEAR) exposed to an electric field to enrich EAMs. The anaerobic slurry of the BEAR was recirculated to the HAD for bio-augmentation with the EAMs. The electric field in the BEAR was demonstrated to enrich the anaerobic digestate with EAMs, and the bio-augmented EAMs significantly enhanced the anaerobic digestion through DIET. Bio-augmentation after the external enrichment of EAMs could be a viable answer to improve a large scale of conventional anaerobic digesters.

## 2. Materials and Methods

### 2.1. Substrate and Inoculum

A mixture of hydrothermally liquefied sludge (HLS) and pulverized food waste (PFW) in an equal volume ratio was used as the substrate for the anaerobic digester. The HLS was prepared by liquifying a waste-activated sludge from a wastewater treatment plant (Incheon, Republic of Korea) at 190 °C for 30 min. The PFW was obtained by collecting food waste from a university cafeteria and grinding it with a household blender (HC-BL2200M, Happy Call Corp., Ltd., Gyeongnam, Republic of Korea). For the initial start-up of the anaerobic digester, an anaerobic digestion sludge was collected from an anaerobic digester in a municipal wastewater reclamation center (B-metro city, Republic of Korea), screened to remove impurities, and then used as the inoculum. Table 1 summarizes the physicochemical properties of the substrate and inoculum.

**Table 1.** Characteristics of hydrothermally liquified sludge (HLS), pulverized food waste (PFW), their mixture used as the feed substrate, and anaerobic digestion sludge used as the inoculum.

| Parameters | HLS | PFW | Mixture | Inoculum |
|---|---|---|---|---|
| pH | 7.64 ± 0.02 | 5.46 ± 0.04 | 5.73 ± 0.04 | 7.58 ± 0.03 |
| Alkalinity (g/L CaCO$_3$) | 12.80 ± 1.00 | 2.60 ± 1.00 | 4.66 ± 0.53 | 3.00 ± 0.10 |
| Total VFAs (g COD/L) | 8.00 ± 0.60 | 1.40 ± 0.40 | 3.30 ± 0.19 | 0.70 ± 0.00 |
| TCOD (g/L) | 64.90 ± 2.00 | 112.10 ± 8.40 | 88.07 ± 6.68 | 12.50 ± 0.00 |
| SCOD (g/L) | 58.50 ± 3.40 | 42.80 ± 7.30 | 47.79 ± 6.09 | 53.50 ± 1.60 |
| TS (g/L) | 58.40 ± 1.70 | 103.60 ± 4.40 | 72.76 ± 1.74 | 118.70 ± 3.00 |
| VS (g/L) | 43.70 ± 1.30 | 87.02 ± 2.50 | 56.27 ± 2.29 | 65.70 ± 2.20 |

### 2.2. Set-Up for the Anaerobic Digestion System and Experimental Design

The lab-scale anaerobic digesters, the HAD and BEAR, were prepared for the experiment. As shown in Figure 1, a horizontal tube (SUS 306, working volume 50 L, diameter 30 cm, and length 120 cm) with closed ends was used as the HAD. A horizontal steel shaft with stirring blades was installed inside the HAD. An electric motor was placed on one side of the reactor to rotate the steel shaft. A wet gas meter (W−NK, Shinagawa Corp., Ltd., Inagi-shi, Japan) was used to monitor biogas production from the HAD. The substrate feeding and biogas venting ports were mounted on the upper side of the HAD. The biogas venting port was connected to the wet gas meter with a rubber tube. Ports for the anaerobic digestate slurry discharge, circulation, and sampling were mounted at the bottom of the HAD. A heating wire (Wooju Electric Heater Co., Ltd., Incheon, Republic of Korea) was wound around the horizontal anaerobic digester to keep the slurry temperature at 35 °C.

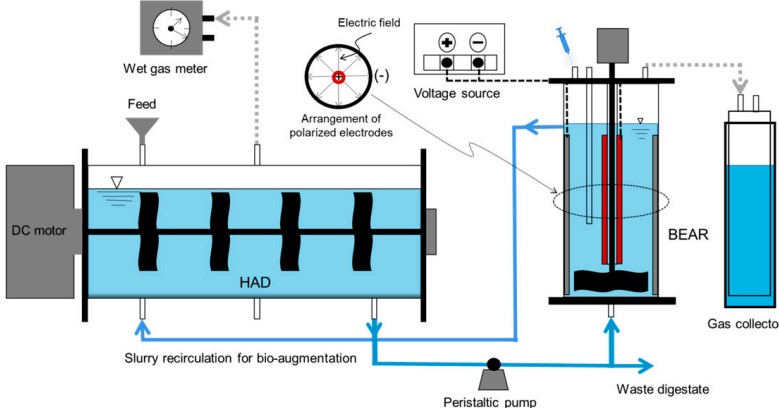

**Figure 1.** Schematic of a horizontal anaerobic digester (HAD) paired with a bioelectrochemical auxiliary reactor (BEAR).

The BEAR was a vertical continuous stirred-tank reactor (CSTR) made of acrylic resin (height 50 cm and diameter 13 cm) with a working volume of 5 L (10% of the HAD). A steel

shaft with a stirring blade was installed in the vertical center of the BEAR. An electric motor for rotating the steel shaft was placed on the cover plate. Ports for biogas sampling and venting and a reference electrode were mounted on the cover plate. The biogas sampling port was sealed with a rubber stopper. The bottom of the reference electrode port was attached with an acrylic tube immersed in the digesting slurry for air tightness. The biogas venting port of the BEAR was connected to a floating-type gas collector using a rubber tube. The gas collector was filled with acidified brine to prevent biogas dissolution. A heating wire was wound around the BEAR to maintain the temperature at 35 °C. Two surface-insulated electrodes were prepared by coating the surface of thin titanium sheets with a polyethylene terephthalate film. The surface-insulated electrodes were installed on the vertical center of the BEAR and the inner wall after rolling the electrodes into an annular shape. The distance of the mounted electrodes between the vertical center of the BEAR and the inner wall was 6 cm.

The HAD was operated in the conventional mode for 430 days before it was paired with the BEAR. After the initial start-up, the flowrate of the feed substrate in the HAD was increased stepwise. From the 211th day, a consistent hydraulic retention time (HRT) of 30 days was established, with exceptions made during the shock loading experiments. On the 431st day, a portion of the anaerobic digestate slurry from the HAD was extracted using a peristaltic pump (Master Flex., Model 7518-10., Vernon Hills, IL, USA) and subsequently introduced to the BEAR, where it was exposed to electric fields to facilitate the enrichment of EAMs. The slurry enriched with EAMs in the BEAR was recirculated to the HAD at rates up to 3.4 L/day for bio-augmentation purposes. Following a pre-established experimental design, the electric field intensities (EFI) reached a maximum of 3.0 V/cm in the BEAR using an external voltage source (DP 30-03TP, Toyotech Co., Incheon, Republic of Korea). The mean exposure time of the digestate slurry to the electric fields in the BEAR, referred to as the Mean Exposure Time (MET), ranged from 1.5 to 30 days, determined by the slurry recirculation rate ($Q_r$) between the HAD and the BEAR (Table 2).

**Table 2.** Experimental conditions with operation time for the HAD and BEAR.

| HAD | | | BEAR | | | |
|---|---|---|---|---|---|---|
| **Day** | **$Q_r$ (L/d)** | **OLR (g COD/L.d)** | **Day** | **MET (d)** | **EFI (V/cm)** | **OLR (g COD/L.d)** |
| 350~ | - | 3.11 ± 0.17 | - | - | - | - |
| 377 | - | 14.24 | - | - | - | - |
| 378~ | - | 3.17 ± 0.18 | - | - | - | - |
| 431~ | 0.17 | 2.85 ± 0.30 | 0 | 30 | 2 | 1.32 ± 0.14 |
| 482~ | 0.85 | 2.57 ± 0.34 | 52 | 6 | 2 | 5.17 ± 0.46 |
| 507~ | 1.7 | 2.59 ± 0.19 | 78 | 3 | 2 | 7.84 ± 0.60 |
| 528~ | 3.4 | 2.52 ± 0.14 | 97 | 1.5 | 2 | 13.91 ± 0.83 |
| 547~ | 1.7 | 2.63 ± 0.17 | 115 | 3 | 1 | 7.26 ± 0.42 |
| 554~ | 1.7 | 2.92 ± 0.03 | 127 | 3 | 0.5 | 7.74 ± 0.53 |
| 559~ | 1.7 | 2.94 ± 0.17 | 136 | 3 | 0 | 8.41 ± 0.11 |
| 564~ | 1.7 | 3.13 ± 0.09 | 148 | 3 | 3 | 7.51 ± 0.63 |

Note: HAD, horizontal anaerobic digester; BEAR, bioelectrochemical auxiliary reactor; Qr bio-augmentation rate; MET, mean exposure time; EFI, electric field intensity; OLR organic loading rate.

### 2.3. Analytical Methods

The pH was measured daily using a pH meter for the anaerobic digestate slurries collected from the HAD and BEAR (MultiLab 4010-3W-YSI, YSI Inc., Yellow Springs, OH, USA). The alkalinity and total VFAs were analyzed using the titration method [3]. HPLC equipped with a UV detector (UltiMate 3000, Dionex, Sunnyvale, CA, USA) was used to analyze the individual VFAs, and the temperature of column (Aminex® 87H column (Bio-Rad, Hercules, CA, USA) was 35 °C. The chemical oxygen demand (COD), total solids (TS), and volatile solids (VS) were analyzed according to the Standard Methods (2005). The daily biogas production rate was monitored using a wet gas meter for the

HAD and a floating-type gas collector for the BEAR. Biogas composition was analyzed with gas chromatography (Gow-Mac Instrument Co., Bethlehem, PA, USA) equipped with a Porapak-Q column (6 ft × 1/8th inch SS) and a thermal conductivity detector. The biogas production rate was converted to a standard pressure and temperature regime by correcting the water vapor pressure at 35 °C [7]. A cyclic voltammogram (CV) for the anaerobic slurry was obtained by scanning the potential range between −1.2 and 0.7 V at 10.0 mV/s using a potentiostat (ZIVE SP1 series, WonATech Co., Ltd., Seoul, Republic of Korea) [5]. A pair of stainless steel meshes (1 cm × 1 cm) were used as working and counter electrodes. The silver–silver chloride electrode (RE-1B Ag/AgCl reference electrode, Tokyo, Japan) was used as the reference electrode. The CV was analyzed to obtain peak potentials for oxidation and reduction and their heights, using Smart Manager (ZIVE BP2 Series, WonATech, Seoul, Republic of Korea).

The bacterial taxonomic profiling was performed of the anaerobic slurry sample collected in a steady state from the BEAR and HAD for 16S rRNA gene-based metagenomic analysis. Following the manufacturer's instructions, one ml of an anaerobic slurry sample was used to extract genomic DNA using DNeasy PowerSoil Pro DNA Kit (QIAGEN, Hilden, Germany). DNA quantification of the extracted DNA was carried out in a QubitTM 4 fluorometer using the QubitTM dsDNA HS Assay Kit (Thermo Fisher Scientific, Waltham, MA 02451, USA). The full-length16S rRNA gene of the extracted genomic DNA from anaerobic slurry samples was amplified through PCR for multiplex sequencing using the 16S barcoding kit SQK-16S024 (Oxford Nanopore Technologies, Oxford Science Park, UK) following the manufacturer's instructions. Sequencing was carried out in an Oxford Nanopore Technologies (ONT) MinION Mk1B device using FLO-MIN106D with R9.4.1 chemistry and the data was collected using super accurate basecalling by the MinKNOW v22.08.9. The generated sequencing data was uploaded to EPI2ME (https://epi2me.nanoporetech.com, accessed on 16 October 2022), a cloud-based analysis platform for sequenced MinION data. The Fastq 16S workflow was used to interpret data using a minimum quality score of 10 for filtering and the read length cut-off between 1000–2000 bp. The Fastq 16S workflow (v2022.01.07) on EPI2ME using NCBI as the reference database revealed the taxonomic classification of base-called reads and their frequency.

### 2.4. Statistical Analysis

The exposure of the digestate slurry to an electric field enriches EAMs in the BEAR, and then the slurry recirculation bio-augments the HAD with the EAMs. So, the electric field intensity (EFI), mean exposure time (MET) to the electric field, and bio-augmentation rate can be the potential manipulated variables for the HAD coupled to the BEAR. The performance of anaerobic digestion was evaluated based on the organic matter removal and methanogenesis. The statistical relationships of the manipulated variables to the anaerobic digestion performance were estimated based on the correlation matrices of Pearson and Spearman. The 'corr' method of the Pandas package in Python was used to obtain the correlation matrices.

In the HAD and BEAR, optimal values of the manipulated variables responding to the methane production rate (MPR) were obtained using the response surface methodology (rsm), employing the 'rsm' package in the statistical software R (4.2.2). The independent variables affecting the MPR ($y$) from the HAD were obtained with the Box–Behnken design for three-factor and three-level. The independent variables were the electric field intensity ($x_1$) of (0–3) V/cm and the circulation rate ($x_2$) of (0–3.4) L/d, and the pH ($x_3$) of (7.19–8.33) was an additional variable to fulfill the number of factors. Coded data were estimated from the experimental data using the function of 'coded.data()' in R. A second-order polynomial equation was used to fit the MPR ($y$) in the HAD as a function of independent variables as follows:

$$y = \beta_0 + \sum_{i=1}^{3} \beta_i x_i + \sum_{i=1}^{3} \beta_{ii} x_{ii}^2 + \sum_{i=1}^{2} \sum_{j=i+1}^{3} \beta_{ij} x_i x_j + \epsilon \tag{1}$$

where $y$ is the dependent variable; $\beta_0$ is a constant; $\beta_i$, $\beta_{ii}$, and $\beta_{ij}$ are the linear, quadratic, and interactive coefficients, respectively; and $\varepsilon$ is the error of the model. The significant differences between independent variables were estimated by fitting the coded data to the second-order equation using the 'rsm' package in R. Non-significant (*p*-value > 0.05) terms were removed from the initial model to achieve a significant model. Experimental data were then refitted to check variations for the fitted model. The fit quality of the model equation was confirmed by the determination coefficient ($R^2$) and the adjusted determination coefficient (adj. $R^2$). The statistical significance of the model was determined by a Fisher test (F-test) based on the p-value with a confidence level of 95%. The correlation coefficient (R), the sum of squares (SS), the mean of squares (MSS), and the F value were also used to analyze the statistical significance of the model. Responses depending on independent variables were visualized with the surface response plot to estimate the optimal conditions of independent variables.

## 3. Results and Discussion

### 3.1. Enrichment of Electroactive Microorganisms

In anaerobic digestion, the enrichment of EAMs and the methanogenesis via DIET can be inferred from distinct features observed in organic matter removal, MPR, methane yield, and methane content in biogas [21,22]. The digestate slurry, the output from the HAD, underwent further degradation of organic residuals in the BEAR, leading to significant biogas production (Figure 2). Typically, the organic residuals in the digestate contain recalcitrant substances, such as lignin, cutin, humic substances, and complex protein compounds [23]. These components are generally resistant to further anaerobic degradation into methane. However, recent studies have highlighted the capability of EAMs to facilitate methane production from stable organic compounds via DIET [3,24,25]. The conversion of organic residuals in the BEAR to methane is likely facilitated by DIET, indicating a significant enrichment of EAMs within the BEAR [13,26].

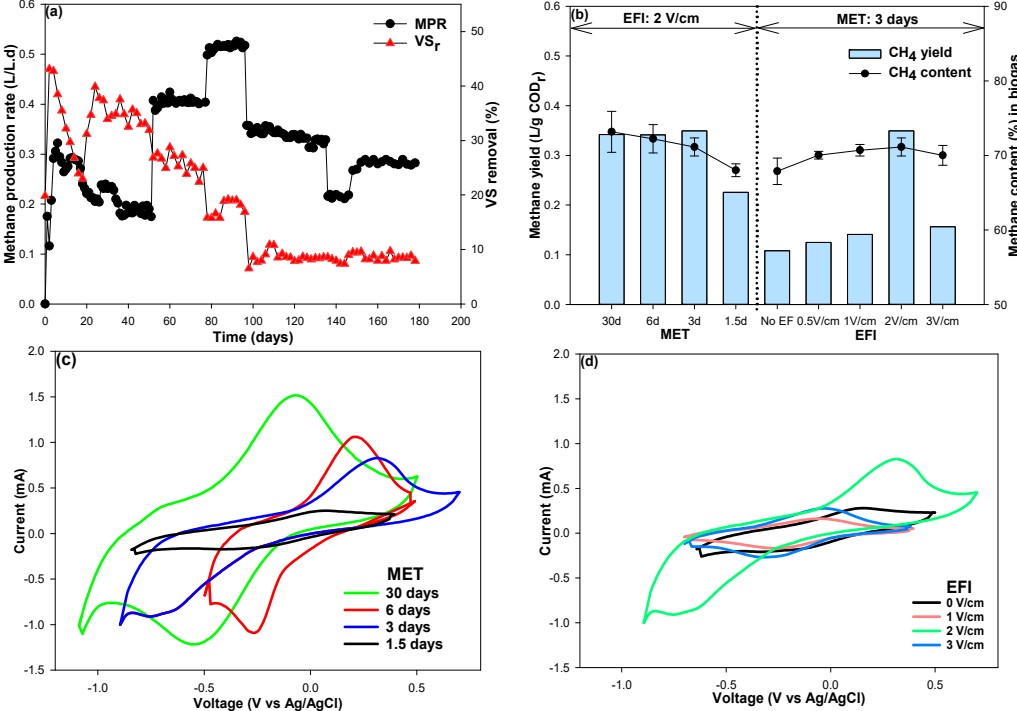

**Figure 2.** Enrichment of EAMs in the BEAR. (**a**) MPR and VS removal, (**b**) methane yield and methane content in biogas, (**c**) cyclic voltammogram with the MET, (**d**) cyclic voltammogram with the EFI (EFI: electric field intensity, MET: mean exposure time, MPR: methane production rate, VS$_r$: volatile solids removal).

However, the methane production from the organic residuals in the digestate was significantly influenced by both the electric field intensity and the mean exposure time of the slurry to the electric fields in the BEAR (Figure 2a). At the electric field intensity of 2 V/cm, VS removal dropped from 33.1% to 8.9% as the mean exposure time decreased from 30 to 1.5 days (Figure 2a). This underscores the crucial role of the exposure duration to the electric field in enriching EAMs in the BEAR. It is worth noting that when paired with the HAD, the BEAR functions as an integrated system. So, the mean exposure time in the BEAR was inversely proportional to the slurry recirculation rate between them. When the exposure time was shortened from 30 to 3 days at 2 V/cm, MPR dramatically increased to 0.52 L/d. Conversely, when the mean exposure time was only 1.5 days, both VS removal and MPR decreased, suggesting that the mean exposure time is insufficient to enrich EAMs adequately. It indicates that the enrichment rate of EAMs is approximately 0.33 $d^{-1}$ at the electric field intensity of 2 V/cm.

Furthermore, the electric field intensity in the BEAR was directly correlated with methane production. For the mean exposure time of 3 days, both VS removal and MPR ascended with increasing electric field intensity, peaking at 2 V/cm and subsequently experiencing a slight decline at 3 V/cm (Figure 2b). Electric fields play an essential role in promoting the growth of electroactive microbes. However, at high electric field intensities, microbial cells might experience stress, potential damage to their cell membranes, and subsequent alterations in their metabolic pathways [27–29]. This result suggests that the optimal electric field intensity to enrich EAMs when using a digestate as the substrate is approximately 2 V/cm.

When the mean exposure time of the slurry to the electric field in the BEAR was longer than 3 days, the methane yield was stable at a high value of 0.34 L/g $COD_r$ under the electric field of 2 V/cm (Figure 2b). The DIET from EEBs to EMAs for methane production conserves more electrons than conventional methanogenesis pathways [5,7]. The stable methane yield, close to the theoretical value of 0.35 L/g $COD_r$, indicates that EMAs were enriched in the BEAR, and the DIET between them was involved in methane production. The molar ratio of methane to carbon dioxide in biogas is mainly influenced by substrate type and carbon dioxide solubility [30,31]. An elevated pH can increase the solubility of carbon dioxide, leading to higher methane content in biogas [5,32]. In addition, a higher methane yield inherently increases the methane content. In the BEAR, the methane content in the biogas was significantly high, ranging from 67.9% to 73.2%, depending on the electric field intensity and mean exposure time of the slurry to the electric field (Figure 2b). These results are also indirect evidence suggesting the enrichment of EAMs and their contribution to methane producton through DIET in the BEAR.

In the BEAR, the enrichment of EAMs was evident from the relative abundance of specific genera in the microbial community. In all conditions exposed to the electric field, the abundant bacteria were Sphingobacterium, Vagococcus, Peptoniphilus, Alkaliphilus, Schaalia, Lactobacillus, and Sedimentibacter in the genus level (Figure 3a). In the previous studies, the genus Sphingobacterium was observed in the biocathode of the microbial fuel cells (MFCs) [33], and the genus Vagococcus, known as a Lactic acid bacteria, was enriched in the biocathode with a temperature change of 10 °C [34,35]. Peptoniphilus is a genus observed in urine-supplied MFCs, and Alkaliphilus and Lactobacillus are potentially EEBs [36,37]. These suggest that a significant majority of the abundant genera were potentially EEBs. However, the relative abundance of these genera varied depending on the electric field intensity and average exposure time. While the genus Sphingobacterium was the most abundant in all conditions exposed to the electric field, the genera Peptoniphilus and Alkaliphilus were notably abundant at the electric field intensity of 2 V/cm and the mean exposure time of 3 days. In control without exposure to the electric field, Vagococcus and Tissierella were abundant, but potential electroactive bacteria such as Sphingobacterium, Peptoniphilus, and Alkaliphilus were found in much lower abundance.

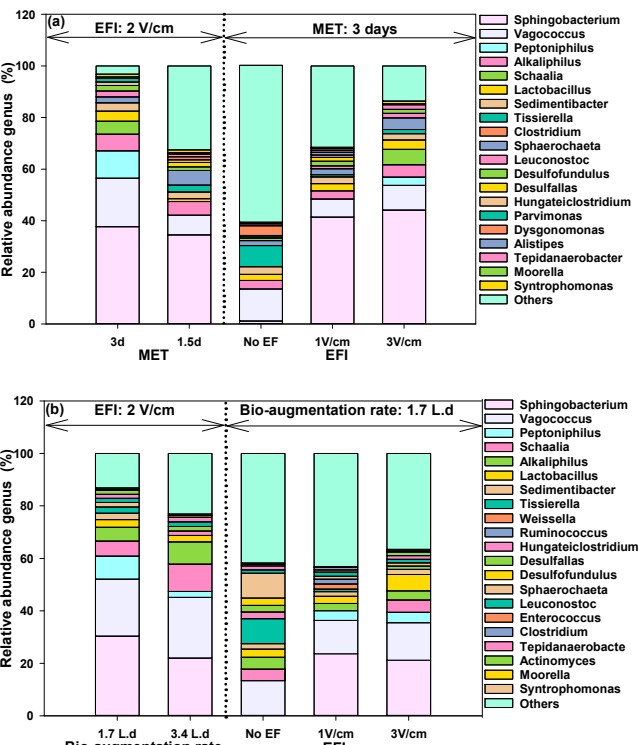

**Figure 3.** Relative abundance of genus. (**a**) BEAR; (**b**) HAD. (EFI: electric field intensity, MET: mean exposure time).

The redox peak height in the CV is another indicator of the activity of EAMs in the slurry and their enrichment within the BEAR [25,38]. Under the electric field intensity of 2 V/cm, there was a decline in the redox peak height when the mean exposure time decreased from 30 to 1.5 days (Figure 2c). At the mean exposure time of 3 days, the redox peak heights were notably high at the electric field intensity of 2 V/cm (Figure 2d). These trends are consistent with the distinct features in MPR, VS removal, methane yield, and methane content in biogas. These results underscore the importance of exposure time to the electric field for the enrichment of EAMs. In control, the genera known as EEBs were also observed, but their electrochemical activity, as determined from the redox peak heights of the CV, was minimal. It suggests that for maintaining their electrochemical activity and facilitating methane production through DIET, EAMs require periodic exposure to an electric field for an adequate exposure time [3,7].

### 3.2. Bio-Augmented Anaerobic Digestion with Electroactive Microorganisms

The bio-augmentation of EAMs significantly improved the anaerobic digestion performance in the HAD and promoted the restoration from its imbalance state after an impulse shock loading. It suggests that the bio-augmented EAMs produce methane through DIET in the HAD without directly applying electric fields or polarized electrodes.

During the conventional mode operation of the HAD, the VS removal reached 56.2%, with an MPR of 0.59 L/L.d and a methane yield of 0.26 L/g COD$_r$ (Figure 4a,b). Furthermore, the biogas had a methane content of 59.9%. These performance values are similar to or slightly better than previous studies on sewage sludge [39], indicating the high biodegradability of the HLS and PFW mixture used as the substrate. However, an impulse shock loading applied on the 377th day led to a significant accumulation of VFA, reaching up to 8.35 g COD/L. The main constituents of the VFA were caproic acid and iso-valeric acid (Figure 4c). This shock event also caused the MPR to plummet to 0.10 L/L.d, coinciding with the VFA increase. Despite a subsequent recovery, the MPR only rebounded to approximately 0.46 L/L.d, still below the pre-shock levels. This behavior suggests that the impulse shock loading severely disrupted the kinetic state of anaerobic digestion. The ensuing imbalance between acidogene-

sis and methanogenesis proved persistent, highlighting a significant limitation of conventional anaerobic digestion regarding resilience [32,40].

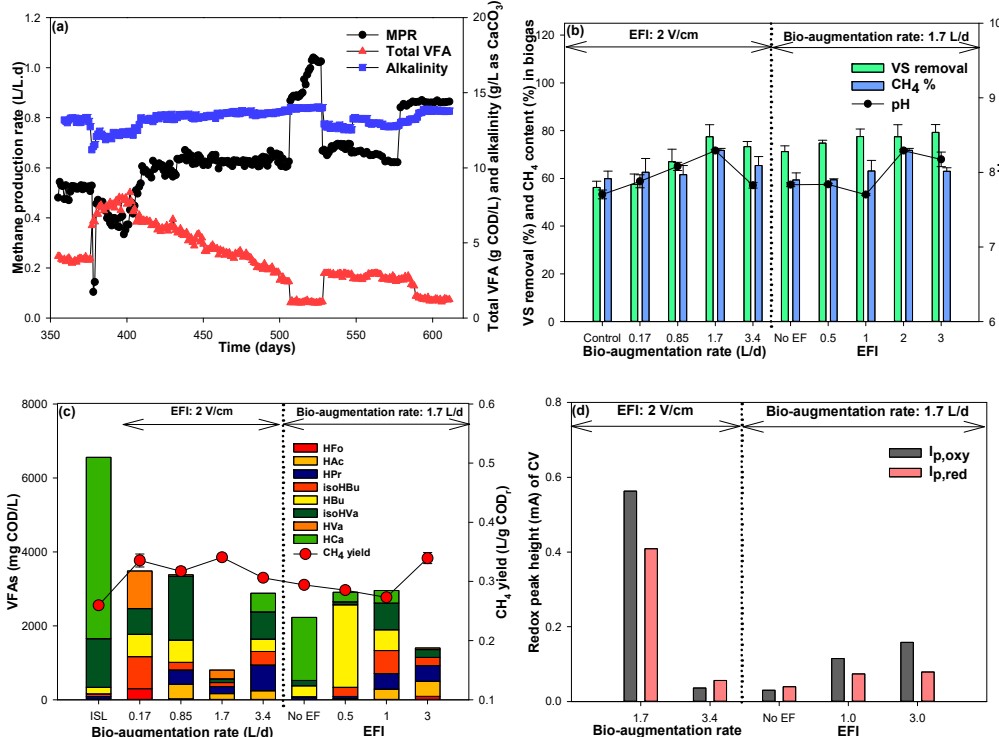

**Figure 4.** State and performance variables in the horizontal anaerobic digester (HAD). (**a**) Methane production rate, alkalinity, and total VFA; (**b**) VS removal, methane content in biogas, and pH; (**c**) VFA composition and methane yield; (**d**) redox peak height (EFI: electric field intensity, ISL: (impulse shock loading), MPR: methane production rate, VFA: volatile fatty acid, VS: volatile solids, $COD_r$: chemical oxygen demand removal, $I_{p,oxy}$: peak height for oxidation, and $I_{p,red}$: peak height for reduction).

On the 431st day, after recirculating the slurry from the BEAR to the HAD for bio-augmentation, there was a notable decrease in total VFA levels, particularly in caproic acid, while methane production increased. These changes were predominantly influenced by the electrochemical activity and recirculation rate of the slurry enriched with EAMs in the BEAR (Figure 4). Specifically, when EAMs were enriched in the BEAR by exposing the slurry to the electric field intensity of 2 V/cm for the mean exposure time of 3 days, a noticeable decrease in VFA was observed in the HAD. Moreover, VS removal impressively reached 77.5%, and the MPR was 1.0 L/L.d, significantly surpassing the conventional mode operation. In addition, the methane yield increased to 0.34 L/g $COD_r$, and methane content in the biogas reached 77.5% (Figure 4b). However, when the bio-augmentation rate through the slurry recirculation from the BEAR was set above 1.7 L/d, drops in VS removal, biogas methane content, and electrochemical activity were observed in the HAD, indicative of potentially insufficient bio-augmentation with EAMs. Intriguingly, even at the bio-augmentation rate of 1.7 L/d from the BEAR, the electrochemical activity identified from the redox peak heights of the CV remained low at electric field intensities other than 2 V/cm (Figure 4d), resulting in diminished methane production, VS removal, and methane content in the biogas. These suggest that, after bio-augmentation, the DIET became the dominant pathway for methane production in the HAD, highlighting the critical role of bio-augmented EAMs in converting VFA into methane [26,41].

The relationship between EAMs, their electrochemical activity, and other prevalent microbial species in the HAD informs that bio-augmenting with EAMs enhances anaerobic digestion performance through DIET [13,32]. In the HAD, the abundant genera in the microbial community were *Sphingobacterium*, *Vagococcus*, *Peptoniphilus*, *Schaalia*, *Alkaliphilus*,

*Lactobacillus*, and *Sedimentibacter* (Figure 3b), and their relative abundance mirrored that of the microbial genera in the BEAR. It suggests that the bioaugmentation with microorganisms enriched in the BEAR plays a crucial role in shaping the microbial community structure of the HAD. In particular, when bio-augmented with microorganisms enriched under an electric field intensity of 2 V/cm for the mean exposure time of 3 days, the relative abundance of these six genera increased, aligning well with the anaerobic digestion performance of HAD. The abundance of the potentially electroactive bacteria, including *Sphingobacterium*, *Vagococcus*, *Peptoniphilus*, *Alkaliphilus*, and *Lactobacillus*, in the microbial community was in good agreement with the redox peak heights in the CV (Figure 4d), and this was sufficient to demonstrate the contribution of DIET to methanogenesis in the HAD. However, among these abundant genera, while *Schaalis*, which belongs to the phylum Actinomycetota, has an as-yet undefined function, *Sedimentibacter* is known to produce acetic acid or butyrate through fermentation [42]. The high anaerobic digestion performance observed in the bio-augmented HAD appears to be a synergistic effect of fermenting and electroactive bacteria.

*3.3. Correlation Analysis for Enrichment of EAMs and Its Impact*

The results from the correlation analysis reveal the intricate dynamics between the electric field and EAMs in the BEAR, and how the bio-augmented EAMs in the HAD subsequently impact anaerobic digestion. Within the BEAR, the role of electric field enriching EAMs can be confirmed by the evident correlations. The redox peak heights for oxidation ($I_{p,oxy}$) and reduction ($I_{p,red}$) in the CV were highly correlated with the mean exposure time, with Pearson's correlation coefficients of 0.63 and 0.79, respectively (Figure 5a) [43]. This underscores the role of the mean exposure time to electric fields in influencing the enrichment of EAMs in the BEAR. The electric field intensity and its interaction with these redox peaks further emphasize the centrality of electric fields in the BEAR. Specifically, Spearman's correlation coefficients of $I_{p,oxy}$ and $I_{p,red}$ with the electric field intensity were 0.42 and 0.60, respectively. This suggests that as the electric field intensity changes, it may directly influence the enrichment of EAMs, impacting their electrochemical activity [3,7]. Redox peak heights showed significant correlations with key indicators in the analysis of anaerobic digestion parameters. Pearson's correlation coefficients between these redox peak heights and parameters like VS and COD removal, methane content in biogas, and methane yield ranged from 0.66 to 0.98. The Spearman's correlation coefficients, which evaluate rank-order relationships, were found between 0.69 and 0.93. It is noteworthy that the Spearman coefficients sometimes exceeded the Pearson coefficients. This indicates that, beyond linear relationships, there might be ranked or ordinal connections between these variables that demand attention.

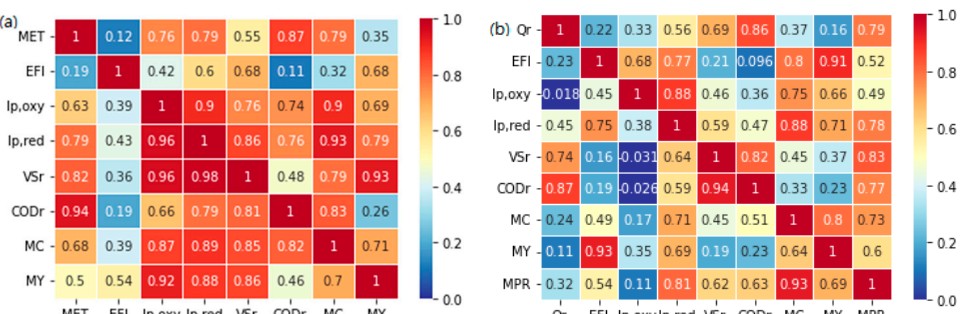

**Figure 5.** Pearson (lower left triangle) and Spearman (upper right triangle) correlation matrices for (**a**) BEAR, (**b**) HAD (BEAR: bioelectrochemical auxiliary reactor, COD$_r$: chemical oxygen demand removal, EFI: electric field intensity, HAD: horizontal anaerobic digester, $I_{p,oxy}$: peak height for oxidation, $I_{p,red}$: peak height for reduction, MC: methane content in biogas, MPR: methane production rate, MY: methane yield, Q$_r$: bio-augmentation rate, MET: mean exposure time to the electric field, VS$_r$: volatile solids removal).

In the HAD, Spearman's correlation coefficients between redox peak heights and the electric field intensity and bio-augmentation rate ranged from 0.33 to 0.77, denoting mild to strong rank-order relationships (Figure 5b). Furthermore, the electric field intensity in the BEAR exhibited notable relationships with key performance metrics, suggesting its direct or indirect influence on the anaerobic digestion performance. What becomes clear from these analyses is the role of bio-augmentation of EAMs recirculating from the BEAR to the HAD. Such correlations indicate a synergistic potential of pairing the BEAR to the HAD, with the electric field in the BEAR playing a pivotal role in optimizing the anaerobic digestion performance in the HAD. The differences between the Spearman and Pearson coefficients presented here enrich our understanding of the BEAR and HAD systems and pave the way for potential refinements and optimization strategies in future investigations.

### 3.4. Optimal Manipulated Variables

The quadratic polynomial equation obtained by fitting the electric field intensity ($x_1$), bio-augmentation rate ($x_2$), and pH ($x_3$) to the MPR in the HAD as the response variable is shown below:

$$y = 1031.669 + 16.536x_1 + 93.275x_2 + 32.332x_3 + 69.074x_1x_3 - 171.633x_1^2 - 281.001x_2^2 - 164.227x_3^2 \qquad (2)$$

In the ANOVA, $p$-values for all model terms except for $x_1$ were less than 0.05 (Table 3). The determinant and adjusted determinant coefficients ($R^2$ and adj. $R^2$) were high at 0.984 and 0.972, respectively. The F value for the model was 79.19, and the $p$-value was close to zero. These results indicate that the quadratic model adequately fits the experimental data. From the response surface plot, the optimal electric field intensity, bio-augmentation rate, and pH for the MPR in the HAD were 2.07 V/cm, 1.98 L/d, and 7.82, respectively (Figure 6). Based on the bio-augmentation rate, the optimal mean exposure time required to enrich EAMs was estimated as 2.53 days for the BEAR.

**Table 3.** Experimental conditions (EFI, HRT, OLR, Qr, and RT) with operation time for the BEAR and HAD.

| Coefficient | Estimates | Std. Error | t-Value | Pr(>\|t\|) |
|:---:|:---:|:---:|:---:|:---:|
| $\beta_0$ | 1031.669 | 16.240 | 63.5269 | $2.994 \times 10^{-13}$ |
| $\beta_1$ | 16.536 | 12.839 | 1.2880 | 0.229885 |
| $\beta_2$ | 93.275 | 12.839 | 7.2651 | $4.739 \times 10^{-5}$ |
| $\beta_3$ | 32.332 | 12.827 | 2.5205 | 0.032742 |
| $\beta_1$: $\beta_3$ | 69.064 | 18.141 | 3.807 | 0.004171 |
| $\beta_{12}$ | −171.632 | 17.697 | −9.6984 | $4.614 \times 10^{-6}$ |
| $\beta_{22}$ | −280.982 | 17.697 | −15.8774 | $6.879 \times 10^{-8}$ |
| $\beta_{32}$ | −164.243 | 17.666 | −9.2972 | $6.540 \times 10^{-6}$ |

[3] Multiple $R^2$: 0.984, Adjusted $R^2$: 0.9716. F-statistic: 79.19 on 7 and 9 DF, $p$-value: $2.309 \times 10^{-7}$.

### 3.5. Implications

The methane production through the DIET of EAMs may be vital to addressing the unresolved intrinsic issues of anaerobic digestion [26,44,45]. The EAMs can be enriched on the surface of polarized electrodes placed in anaerobic digesters [8,46,47]. However, methane production through DIET becomes meaningful in improving anaerobic digestion only in the anaerobic digester with polarized electrodes of sufficient area. However, a large budget is initially required to install polarized electrodes of sufficient area in an anaerobic digester [7]. In addition, there are severe issues with the maintenance of the electrode, such as replacing the deteriorated electrodes, and the installed electrodes in the anaerobic digester may interfere with the agitation of the digesting slurry and the cleaning work [7,31]. Therefore, the enriched EAMs on the surface of polarized electrodes have not yet been applied to improve the anaerobic digestion in on-site scale facilities. Fortunately, in bioelectrochemical reactors, an electric field can enrich the bulk solution with EAMs [7,48]. In addition, only a relatively small size of polarized electrodes is sufficient to form the

electric field necessary to enrich EAMs. The electrode lifetime can also be extended by insulating the surface with a durable dielectric polymer [5].

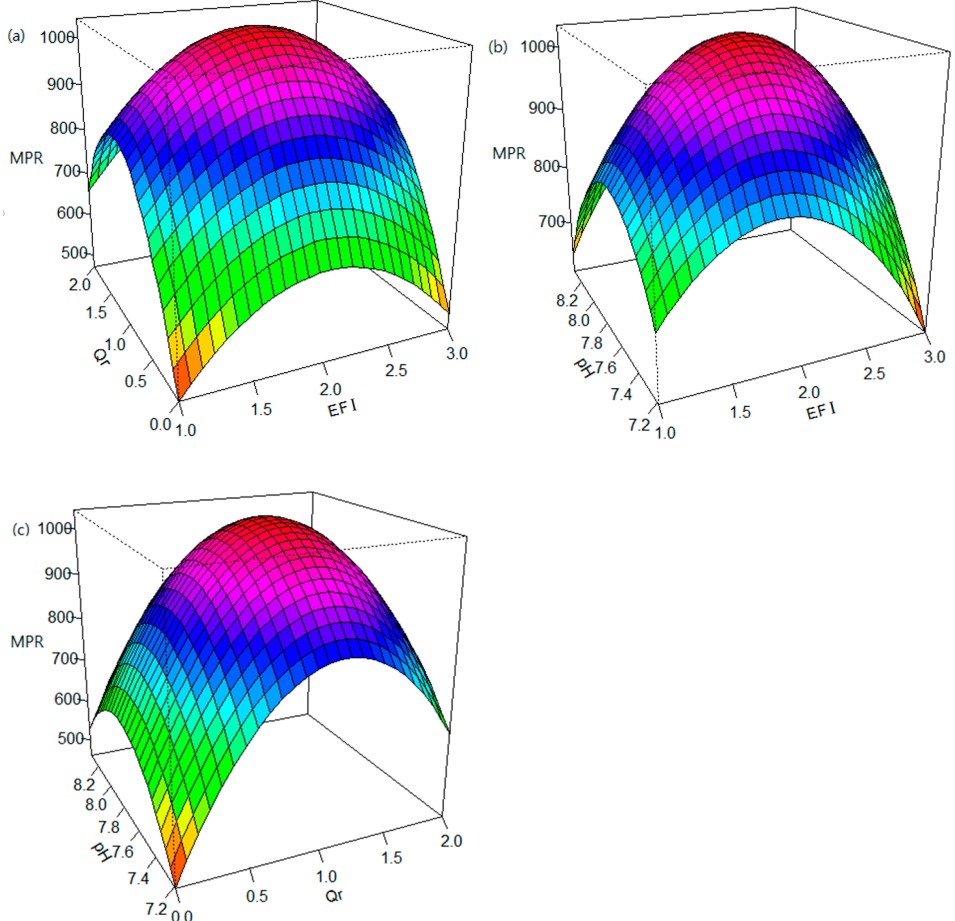

**Figure 6.** Response surface plot of methane production rate in the HAD for (**a**) bio-augmentation rate and electric field intensity, (**b**) pH and electric field intensity, and (**c**) pH and bio-augmentation rate (BEAR: bioelectrochemical auxiliary reactor, EFI: electric field intensity, HAD: horizontal anaerobic digester, MPR: methane production rate, Qr: bio-augmentation rate through the slurry circulation).

Therefore, the bio-augmentation after the enrichment of EAMs with the electric field in a small external bioelectrochemical reactor could significantly alleviate the issues related to installing and maintaining polarized electrodes in anaerobic digesters. However, few studies have improved anaerobic digestion through bio-augmentation after enriching EAMs in a small external bioelectrochemical reactor. This study found that an electric field in the BEAR enriches the anaerobic digestate, discharged from the HAD, with EAMs and that bio-augmenting the HAD with the enriched EAMs significantly improves anaerobic digestion performance. The electric field intensity and mean exposure time to the electric field were the manipulated variables in the BEAR for enriching EAMs. The optimal values estimated from the response surface methodology were 2.07 V/cm for the electric field intensity and 2.53 days for the mean exposure time (Figure 6). The bio-augmentation to the HAD with the enriched EAMs accelerated the conversion of VFAs to methane through DIET, which improved anaerobic digestion by increasing pH and alkalinity and promoted the restoration of the anaerobic digestion state from the kinetic imbalance. During the operation of the HAD in the conventional mode, the VS removal was 56.2%, and the MPR was 0.59 L $CH_4$/L.d (Figure 3). However, the methane content in biogas and the methane yield were low, with 59.9% and 0.26 L/g COD removed, respectively. However, at the optimal conditions, the bio-augmentation to the HAD significantly improved the VS

removal to 79.1%, the MPR to 1.00 L $CH_4$/L.d, the methane content in biogas to 71.6%, and the methane yield to 0.34 L/g $COD_r$.

The BEAR volume for enriching EAMs was only 10% of the HAD, but as well as the HAD, the additional operation of the BEAR can still be technically and economically burdensome. In particular, additional energy for heating, stirring, and slurry pumping is required to operate the BEAR. However, such a burden could be alleviated by reducing the BEAR volume by optimizing the enrichment processes for EAMs [3,7,47]. These findings imply that coupling the HAD with the BEAR for bio-augmenting EAMs after external enrichment could be a viable way to apply DIET to anaerobic digestion.

## 4. Conclusions

Electric fields can enrich EAMs, which can be electrically coupled to electrotrophic methanogenic archaea to enhance anaerobic digestion via DIET. The optimal conditions for enriching EAMs in the bioelectrochemical auxiliary reactor are the electric field intensity of 2.07 V/cm and the mean exposure time of 2.53 d to the electric field. The bio-augmented EAMs actively produce methane through DIET and help restore a kinetic imbalance of anaerobic digestion. The bio-augmentation improves the VS removal to 79.1%, MPR to 1.00 L/L.d, methane yield to 0.34 L/g COD removed, and biogas methane content to 71.6%. The bio-augmentation with externally enriched EAMs could be a way to apply the DIET of EAMs to improve anaerobic digestion.

**Author Contributions:** Investigation, data curation, Writing—original draft, visualization, Z.-K.A.; conceptualization, methodology, resources, data curation, writing—review and editing, visualization, Y.-C.S.; conceptualization, methodology, K.-T.K.; conceptualization, methodology, C.-Y.L.; data curation, S.-H.J.; supervision, B.-U.B. All authors have read and agreed to the published version of the manuscript.

**Funding:** This research was mainly supported by a grant (NRF-2022R1A2C1009440) of the National Research Foundation of Korea (NRF), funded by the Korea government (MIST) and partly by the Ministry of Land, Infrastructure, and Transport of the Korea government (23UGCP-B157945-03).

**Institutional Review Board Statement:** Not applicable.

**Informed Consent Statement:** Not applicable.

**Data Availability Statement:** Data is contained within the article.

**Acknowledgments:** This work was supported by 2023 BK21 FOUR Graduate School Innovation Support by Korea Maritime & Ocean University ([KMOU Graduate School Innovation]—Fellowship Program).

**Conflicts of Interest:** The authors declare no conflict of interest.

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
