# Peer review of "The Bioaugmentation of Electroactive Microorganisms Enhances Anaerobic Digestion"

_fermentation, doi:10.3390/fermentation9110988_

Round 1

Reviewer 1 Report

Comments and Suggestions for Authors

Dear Authors,

Thank you so much for the very interesting work. The authors have investigated a very crucial topic and they have produced some very interesting results. Thus, I would recommend this paper to be considered for acceptance after some minor revisions.

While the authors have done a very good job in preparing the manuscript, it is recommended that the authors must consider the follow points or suggestions below:

1. Title of the paper: The title should be modified.

2. Highlights: Authors must provide the highlights to the paper.

3. Introduction section: Authors are strongly suggested tp expand their literature survey, so that they should be able to identify the research gap, that will inform their study. 

4. Fig. 5 must be expanded. They are too small and not readable.

5. English language application: Authors should proofread the entire document again and check for spelling errors.

All the best of luck to your revision. It is indeed a brilliant paper!

Comments on the Quality of English Language

English language application: Authors should proofread the entire document again and check for spelling errors.

Author Response

The authors would like to thank the reviewer for the valuable comments. The manuscript was carefully revised based on the comments, and all changes were highlighted in the revised version as follows.

We very much hope the revised manuscript is accepted for publication in Fermentation.

Response to reviewer's comments

Manuscript: fermentation-2660940

Title: Bioaugmentation of externally enriched electroactive microorganisms enhances anaerobic digestion

Author: Zheng-Kai An, Young-Chae Song, Keug-Tae Kim, Chae-Young Lee, Seong-Ho Jang, Byung-Uk Bae

The authors would like to thank the reviewer for the valuable comments. The manuscript was carefully revised based on the comments, and all changes were highlighted in the revised version as follows.

We very much hope the revised manuscript is accepted for publication in Fermentation.

Young-Chae Song, Ph.D., PE, Professor

Department of Environmental Engineering

Korea Maritime and Ocean University

Dear Authors, Thank you so much for the very interesting work. The authors have investigated a very crucial topic and they have produced some very interesting results. Thus, I would recommend this paper to be considered for acceptance after some minor revisions.

While the authors have done a very good job in preparing the manuscript, it is recommended that the authors must consider the follow points or suggestions below:

  1. Title of the paper: The title should be modified.
    Ans) The authors sincerely thank the reviewer for the insightful feedback. The title has been revised for improved clarity in response to the reviewer's suggestions. The original title, "Bioaugmentation of externally enriched electroactive microorganisms enhances anaerobic digestion," was amended to "Bioaugmentation of electroactive microorganisms enhances anaerobic digestion" to better understand the manuscript's content.
  2. Highlights: Authors must provide the highlights to the paper.
    Ans) The authors express their gratitude to the reviewer for the comments. The authors realize that Highlights are a key part and their importance to the paper. Based on the reviewer's suggestion, HIGHLIGHTS have been added as follows.

Highlights:

- An electric field enriches anaerobic digestate with electroactive microorganisms (EAMs).

- The optimal electric field intensity and mean exposure time to enrich EAMs are 2.07 V/cm and 2.53 d, respectively. - Bio-augmented EAMs in anaerobic digestion promote methanogenesis through DIET.

- The bio-augmentation of EAMs enhances the resilience of anaerobic digestion against imbalance.

- EAMs can be applied to anaerobic digestion through external enrichment and bio-augmentation.

 (Highlighted in lines 34- 40 on page 1).

  1. Introduction section: Authors are strongly suggested to expand their literature survey, so that they should be able to identify the research gap, that will inform their study. 

Ans) The authors sincerely appreciate the valuable comments. We have checked the literature carefully and added the current application status and technical barriers into the INTRODUCTION part of the revised manuscript to explain the research gap. The new additions are as follows.

"Nevertheless, several challenges related to electrode installation and maintenance have hindered the practical application of MECs in full-scale anaerobic digesters [7, 17, 34]." (Highlighted in lines 74- 76 on page 2).

"Fortunately, it has been discovered that an electric field can enrich EAMs within the bulk solution of anaerobic digesters [7, 18]. ...... This study proposes that EAMs could be enriched in an external small bioelectrochemical reactor without the severe issues associated with the electrode and bio-augmented to improve anaerobic digestion." (Highlighted in lines 79- 93 on page 2).

  1. Fig. 5 must be expanded. They are too small and not readable.
    Ans) The authors have taken into account the reviewer's feedback, and as a result, the original Fig.5, which has been renumbered as Fig.6, has been resized for better clarity and is now legible (lines 451- 453 on page 12)

  1. English language application: Authors should proofread the entire document again and check for spelling errors.
    ANS) We have proofread the entire manuscript again and corrected any spelling errors found.

Reviewer 2 Report

Comments and Suggestions for Authors

The article by Zheng-Kai An et al. is interesting in that it describes a long-term experiment, lasting over a year, on the influence of an electric field on the increase in electroactive microorganisms and the production of methane in a mixture of HLS and PFW. The fact that the work was carried out over a long period of time using reactors of an appropriate size is to be commended.

However, the article suffers from two major problems. The first is that the authors have tested multiple parameters that interfere with each other, and the least that can be said is that the article needs to be completely rewritten to explain the results. For instance, Figures 2 and 3, and the text describing the curves and results, are too crowded, illegible and even incomprehensible.

The second problem lies in the argument and the main philosophy of the article: the authors claim that the application of an electric field enriches the medium in EAMs and that, as a result, the other significant parameters of methanisation are enhanced. However, the results shown in Figure 2 and 3 do not show this enrichment, especially since, on the one hand, the uncertainties of the results are not given and are generally high and, on the other hand, these organisms only represent about 2.4% in the BEAR (line 270) and 5% in the HAD (line 341). Moreover, the parameters introduced, the Pearson and Spearman matrices, are parameters that are influenced by the electric fields involved, without any enrichment being demonstrated here.

In summary, the reviewer believes that the basis of the article needs to be reviewed by demonstrating the effects of electric fields on methanogenic activity and deducing that one of the reasons could be an enrichment of certain microorganisms.

Furthermore, could the authors explain why the BEAR's operating parameters are worse with a field of 3 V/cm than with 2 V/cm for the same residence time of 3 days?

Author Response

The authors would like to thank the reviewer for the valuable comments. The manuscript was carefully revised based on the comments, and all changes were highlighted in the revised version as follows.

We very much hope the revised manuscript is accepted for publication in Fermentation.

Response to reviewer's comments

Manuscript: fermentation-2660940

Title: Bioaugmentation of externally enriched electroactive microorganisms enhances anaerobic digestion

Author: Zheng-Kai An, Young-Chae Song, Keug-Tae Kim, Chae-Young Lee, Seong-Ho Jang, Byung-Uk Bae

The authors would like to thank the reviewer for the valuable comments. The manuscript was carefully revised based on the comments, and all changes were highlighted in the revised version as follows.

We very much hope the revised manuscript is accepted for publication in Fermentation.

Young-Chae Song, Ph.D., PE, Professor

Department of Environmental Engineering

Korea Maritime and Ocean University

  • Comments from the second reviewer

The article by Zheng-Kai An et al. is interesting in that it describes a long-term experiment, lasting over a year, on the influence of an electric field on the increase in electroactive microorganisms and the production of methane in a mixture of HLS and PFW. The fact that the work was carried out over a long period of time using reactors of an appropriate size is to be commended.

  • However, the article suffers from two major problems. The first is that the authors have tested multiple parameters that interfere with each other, and the least that can be said is that the article needs to be completely rewritten to explain the results. For instance, Figures 2 and 3, and the text describing the curves and results, are too crowded, illegible and even incomprehensible.

Ans) We appreciate the reviewer's comments, which have allowed us to re-examine and reinterpret the data from our intriguing findings in the submitted manuscript. The original explanation regarding the impact of electric field intensity and exposure time to the electric field to enrich EAMs and the enhanced anaerobic digestion performance in the HAD by bio-augmentation was somewhat complex and challenging to understand. To make our significant findings more accessible to readers, we have analyzed the raw data and re-drawn Figures 2 and 3 as figures 2, 3, and 4. Based on the reviewer's feedback, we have completely rewritten the sections "3.1 Enrichment of electroactive microorganisms" and "3.2 Bio-augmented anaerobic digestion with electroactive microorganisms," as highlighted on pages 6-10 in the revised manuscript.

  • The second problem lies in the argument and the main philosophy of the article: the authors claim that the application of an electric field enriches the medium in EAMs and that, as a result, the other significant parameters of methanisation are enhanced. However, the results shown in Figure 2 and 3 do not show this enrichment, especially since, on the one hand, the uncertainties of the results are not given and are generally high and, on the other hand, these organisms only represent about 2.4% in the BEAR (line 270) and 5% in the HAD (line 341).

Ans) We are grateful for the insightful comments about the results showing that when the BEAR was used to enrich EAMs using an electric field, which was then used to bio-augment the HAD, there was a notable enhancement in anaerobic digestion performance. The digestate residues discharged from the HAD operated at an HRT of 30 days are generally challenging to degrade further anaerobically. However, while enriching EAMs in the BEAR by applying an electric field, we observed additional VS removal, significant methane generation, and high methane yield and methane content in the biogas from the anaerobically digested residues discharged from the HAD. Such phenomena are hard to observe in conventional anaerobic digestion and are similar to results typically seen when EAMs produce methane through DIET. Although the submitted manuscript described that the abundance of electroactive bacteria at the species level was only 2.4%, this figure includes only known electroactive species. Since known electroactive species at the species level are limited, and it is challenging to include all species with low abundance, we re-checked at the genus level in the revised paper. At this point, we observed a remarkable result where 5 out of the top 7 abundant genera in the microbial community of the BEAR are potential electroactive bacteria (Figure 3 in the revised manuscript). Additionally, the redox peaks in the CV provide more direct evidence that the electric field enriched EAMs in the BEAR (Figure 2c & d and Figure 4 in the revised manuscript).

Notably, the kinetic imbalance caused by an impulse shock loading improved rapidly after pairing the HAD with the BEAR. The organic matter reduction, methane generation rate, methane yield, and methane content in biogas were significantly higher than before the pairing. Such outcomes were highly reminiscent of the results seen when DIET mediated by EAMs contributes to methane production. Additionally, pairing the HAD with the BEAR signifies the bio-augmentation of the HAD with EAMs enriched in the BEAR. After the bio-augmentation, the microbial community of the HAD at the genus level resembled that of the BEAR considerably, and distinct redox peaks were observed in the CV. Following the bio-augmentation of the HAD by pairing with the BEAR, the tracers of DIET were evident in the enhanced organic matter degradation, increased methane production rate, methane yield, and elevated methane content in biogas.

Additionally, a significant proportion of potential electroactive bacteria and distinct redox peaks underscore this observation in the HAD. This highlights the positive impact of the bio-augmented EAMs on the anaerobic digestion performance. These details have been articulated in the newly written sections 3.1 and 3.2 (Highlighted on pages 6-10 in the revised manuscript).

  • Moreover, the parameters introduced, the Pearson and Spearman matrices, are parameters that are influenced by the electric fields involved, without any enrichment being demonstrated here.

Ans) Thank you for the reviewer's comments on the correlation between parameters and electroactive In the revised manuscript, taking into account the feedback from the comments, we have entirely rewritten the content to focus on the correlation between the electric field and average exposure time in the BEAR and the redox peak heights of the CV representing the enrichment and activity of EAMs, and how bio-augmentation with EAMs in the HAD is correlated with improvements in anaerobic digestion performance. Additionally, to provide a more detailed explanation about the interaction between the electric field and EAMs, we have removed the section on Granger causality and placed a stronger emphasis on correlation analysis, including a discussion on the differences between Pearson and Spearman correlation coefficients (Highlighted on page 10-11).

  • In summary, the reviewer believes that the basis of the article needs to be reviewed by demonstrating the effects of electric fields on methanogenic activity and deducing that one of the reasons could be an enrichment of certain microorganisms.

Ans) As mentioned in the comments, in the revised manuscript, sections 3.1 to 3.3 were revisited to offer readers a more precise insight that electric fields can enhance EAMs and that recirculating these bio-augmented EAMs into the HAD can notably boost anaerobic digestion performance. The raw data was thoroughly reanalyzed and rewritten.

  • Furthermore, could the authors explain why the BEAR's operating parameters are worse with a field of 3 V/cm than with 2 V/cm for the same residence time of 3 days?

Ans) Thank you for raising this specific concern about the operating parameters of BEAR in different electric field intensities. To address your concerns regarding the difference in performance at 3 V/cm compared to 2 V/cm for the mean exposure time of 3 days: Higher electric field intensities, while they may seem advantageous, can sometimes have unintended consequences on the microbial community. Specifically, a more intense field tends to select specific EAMs, reducing the diversity within the community. On the other hand, a lower intensity can foster a greater diversity of EAM species. It is a misconception that a stronger electric field will always lead to the enrichment of more EAMs. In fact, at high electric field intensities, microbial cells might experience stress, potential damage to their cell membranes, and subsequent alterations in their metabolic pathways. We have further elaborated on the relationship between electric field intensity and the enrichment of EAMs in the revised manuscript. (Highlighted in lines 267- 272 on page 7).